# Interaction between Gut Microbiota and Celiac Disease: From Pathogenesis to Treatment

**DOI:** 10.3390/cells12060823

**Published:** 2023-03-07

**Authors:** Roberta Elisa Rossi, Giulia Dispinzieri, Alessandra Elvevi, Sara Massironi

**Affiliations:** 1Gastroenterology and Endoscopy Unit, IRCCS Humanitas Research Hospital, Via Manzoni 56, Rozzano, 20089 Milan, Italy; 2Division of Gastroenterology and Center for Autoimmune Liver Diseases, Department of Medicine and Surgery, University of Milano-Bicocca and European Reference Network on Hepatological Diseases (ERN RARE-LIVER), San Gerardo Hospital, Via Pergolesi 33, 20900 Monza, Italy

**Keywords:** intestinal microbiota, gut microbiota, celiac disease, immune response

## Abstract

Celiac disease (CD) is a common systemic disorder that results from an abnormal response of human immunity to gluten intake, affecting the small intestine. In individuals who carry a genetic susceptibility, CD is triggered by environmental factors, including viral infections and dysbiosis of the gut microbiota. The gut microbiome is essential in controlling the immune system, and recent findings indicate that changes in the gut microbiome may contribute to various chronic immune disorders, such as CD through mechanisms that still require further exploration. Some bacteria exhibit epitopes that mimic gliadin and may enhance an immune response in the host. Other bacteria, including *Pseudomonas aeruginosa*, may work in conjunction with gluten to trigger and escalate intestinal inflammation. The microbiota may also directly influence antigen development through the production of immunogenic or tolerogenic gluten peptides or directly influence intestinal permeability through the release of zonulin. Finally, the gut microbiome can impact intestinal inflammation by generating proinflammatory or anti-inflammatory cytokines and metabolites. It is crucial to consider the impact of genetic factors (specifically, HLA-DQ haplotypes), perinatal elements such as birth mode, type of infant feeding, and antibiotic and infection exposure on the composition of the early intestinal microbiome. According to the available studies, the gut microbiome alterations associated with CD tend to exhibit a decreased presence of beneficial bacteria, including some anti-inflammatory *Bifidobacterium* species. However, some controversy remains as some reports have found no significant differences between the gut microbiomes of individuals with and without CD. A better understanding of the gut microbiome’s role in the development of CD would greatly benefit both prevention and treatment efforts, especially in complicated or treatment-resistant cases. Here, we have attempted to summarize the available evidence on the relationship between the gut microbiota and CD, with a particular focus on potential therapeutic targets.

## 1. Introduction

Celiac Disease (CD) is a systemic immune disease affecting the small intestine in individuals with genetic susceptibility. It is determined by the consumption of gluten, a protein found in grains including wheat, barley, rye, spelt, and kamut that is soluble in alcohol. CD is a common disease with a prevalence that varies by age, sex, and geographic region. The global seroprevalence of CD is approximatelly 1.4% while the global prevalence of CD, confirmed by biopsy, being 0.7%, with the highest prevalence in Europe and Oceania (0.8%) and the lowest prevalence in South America (0.4%) [1]. The accuracy of seroprevalence may be flawed due to the nonspecific nature of serology, while the prevalence determined by biopsy may be underestimated as not all individuals are willing to undergo endoscopic examination. CD is 1.5 times more common in women than in men and is more often diagnosed in children but can occur at any age, including the elderly. The prevalence is higher among individuals with a first-degree relative who has CD (10% to 15%), Hashimoto’s thyroiditis (5%), type 1 diabetes (3% to 16%), or other autoimmune diseases (such as autoimmune liver disease, Sjogren’s syndrome, and IgA nephropathy), Down syndrome (5%), Turner syndrome (3%), and IgA deficiency (9%) [2]. Over the past 50 years, the incidence of CD has increased, due in part to better diagnostic tools and more thorough screening of people at high risk of disease. However, the rise in CD incidence is not only due to improved detection but also to a real increase in the disease itself. The exact reasons behind this trend are not yet fully understood, but potential contributing factors, such as changes in environmental factors may lead to loss of gluten tolerance in the diet [3].

### 1.1. Pathogenesis

The pathogenesis of CD involves a complex interplay of genetic and environmental factors.

Genetic predisposition and exposure to gluten may trigger an innate gut pro-inflammatory response and an inappropriate adaptive response due to loss of gluten tolerance. A strong genetic predisposition is associated with the presence of human leukocyte antigen (HLA) risk alleles. Haplotypes DQ2 and DQ8, expressed on the surface of antigen-presenting cells, can bind activated (deamidated) gluten peptides and elicit an abnormal immune response. The DQ2 and DQ8 haplotypes are necessary but not sufficient for the development of CD [2]. Despite the high prevalence of HLADQ2/HLA-DQ8 in the general population (25–35%), only a small portion (3%) of these individuals with the HLA gene develop CD. This highlights the complexity of CD pathogenesis, and genome-wide association studies have identified over 100 non-HLA genes contributing to the disease [4]. Exposure to gluten is essential for the development of CD. However, factors such as duration of breastfeeding or timing of gluten introduction do not impact its risk. Gluten intolerance can be triggered by various causes, including gastrointestinal infections, medications, alpha-interferon, and surgical procedures, at any point in a person’s life [5]. Gliadin, a main component of gluten, is partially broken down into peptides. These peptides trigger the release of zonulin, which can dissolve tight junctions, making the intestine more permeable. Gluten can pass through this opening and enter the lamina propria via a paracellular mechanism. It has been observed that gluten can cross the intestinal barrier via the transcellular pathway once tolerance to gluten is broken. The transferrin receptor CD71 is usually expressed on the basolateral side of enterocytes. However, in people with celiac disease (CD) during the active phase of the disease, it is over-expressed on the luminal side of the intestinal epithelium. This leads to the retrotranscytosis of gliadin peptides, combined with the secretory IgA, from the apical to the basal surface of the cells. This retrotranscytosis of secretory IgA-gliadin shields gluten fragments from destruction by lysosomes, and instead allows the harmful gluten peptides to enter the lamina propria of the intestine. This perpetuates the intestinal inflammation triggered by the initial passage of the peptides through the spaces between the cells [4]. 

The adaptive immune response to gluten, especially gliadin, involves antigen-presenting cells such as macrophages, dendritic cells, and B cells that carry HLA class II DQ2 or DQ8 molecules on their surface, which are deputed for the uptakeand presentation of gliadin peptides. Indeed, these cells recognize and internalize the gluten peptides, then present them to CD41 TH1 cells, which specifically react to gliadin, by releasing inflammatory cytokines, such as IFN-g and thus leading to a cascade of immune responses [6]. Therefore, the paracellular and/or transcellular passage of gliadin peptides through the intestinal barrier triggers an immune response, leading to inflammation in the small intestine. The combination of these inflammatory events results in damage to the intestinal mucosa with villous atrophy and crypt hyperplasia, resulting in malabsorption and nutrient deficiencies.

The gut microbiota is also thought to play a role in the pathogenesis of CD, which will be discussed later.

### 1.2. The Role of Gut Microbiota in the Pathogenesis of Celiac Disease

There is increasing evidence of the role of the gut microbiota in the pathogenesis of CD, which can be explained in part as follows: first, the prevalence of CD has increased rapidly over the past two decades, and this cannot be attributed solely to a genetic basis [7]; only few genetically predisposed individuals develop an active disease [8], again highlighting the role of external factors; moreover, delivery by cesarean section, that highly affects the microbial composition of the intestine of newborns [9], has been associated with increased susceptibility to CD [10]. Finally, a positive correlation between early antibiotic use and the development of CD was found [11], even though with conflicting results so far. 

Indeed, research has suggested that alterations in the composition and function of the gut microbiome can contribute to the development and progression of CD, through different but not yet fully elucidated mechanisms. 

For instance, several studies focused on the duodenal and fecal microbiota in order to possibly find a relationship between the occurrence of CD and the gut microbiota. Alterations in the microbiota associated with CD include lower abundance of beneficial bacteria within the microbiome, particularly *Bifidobacterium* species which are known for their anti-inflammatory and immunomodulatory properties [12,13,14,15]. *Lactobacillus* species are also thought to be beneficial, through several mechanisms including secretion of anti-inflammatory cytokines and alteration of the Th1 immune response [16,17]. 

On the other hand, the overgrowth of certain bacteria has been associated with increased intestinal permeability, which is a hallmark of CD. Specifically, some *Bacteroides* species have been described in higher aboundance in CD patients, which may contribute to the degradation of mucins and increased intestinal permeability [15,18]. Similarly, some studies reported an increased prevalence of *Escherichia coli* in patients with CD both active and in remission compared with healthy controls [15,18] as well as a higher abundance of specific *Staphylococcus* species [15,19]. 

Regarding the mechanisms by which the gut microbiota may contribute to CD pathogenesis, there are several hypotheses (Table 1). One hypothesis is that some gut bacteria express epitopes that mimic gliadin, a component of gluten, which can elicit a host immune response. This response leads to the activation of the immune system and the production of antibodies that attack the gut lining [20]. Another hypothesis is related to lipopolysaccharides, molecules found in Gram-negative bacteria’s outer membrane. Lipopolysaccharides can play a relevant role in both the innate and adaptive immune systems through the production of interleukin-15 (IL-15) which can trigger inflammation in the gut [21]. In addition, some bacteria, such as *Pseudomonas aeruginosa*, can cause increased inflammation of the mucous membranes in combination with gluten [22]. The combination of these two factors can lead to further damage to the gut lining and an exacerbation of CD symptoms. Finally, viral infections can also act as triggers for the activation of the innate immune system. In detail, Toll-like receptor 3 (TLR3) is a receptor that can recognize and respond to viral infections, leading to activation of the innate immune system and intestinal inflammation [23]. These mechanisms highlight the complex interplay between the gut microbiome, gluten, and the immune system in the development of CD. 

On the other hand, the microbiota itself may contribute to CD pathogenesis by altering the digestion of gluten through he production of specific immunogenic or tolerogenic gluten peptides, by directly releasing zonulin which affects intestinal permeability, and by promoting epithelial mucosal maturation. The production of proinflammatory or anti-inflammatory peptides and cytokines by the gut microbiota could also contribute through the modulation of the immune system [24].

Gut bacteria can also regulate intestinal barrier function and the immune response to dietary antigens through the release of short-chain fatty acids (SCFAs). SCFAs, including acetate, propionate, and butyrate, are produced by the fermentation of dietary fibers by gut bacteria. These SCFAs are involved in maintaining the integrity of the intestinal epithelial barrier by promoting the tight junction formation, inhibiting pro-inflammatory cytokine production, and promoting regulatory T-cell differentiation. In this way, the balance of gut bacteria and their by-products, such as SCFAs, can affect the host’s ability to tolerate dietary antigens, including gluten, possibly leading to the development of CD. Recently it has been demonstrated that free circulating fatty acids (comprising SCFAs) of patients with celiac disease (CD) showed a different composition compared to healthy controls, with a strong positive association between CD and butyric acid, thus suggesting a disease-specific microbiota [42].

It is well known that CD occurs in genetically predisposed patients and that the expression of HLA-DQ haplotypes has been recognized as a necessary but not sufficient element for the development of the disease. It is important to highlight that, according to several prospective data, HLA-DQ haplotypes affect the composition of the early gut microbiota. Olivares et al. [25], in their prospective study of 22 exclusively breastfed and vaginally delivered children who were either at high genetic risk (HLA-DQ2 carriers) or low genetic risk (non-HLA-DQ2/8 carriers) for CD occurrence, found that the genotype of infants at familial risk for CD, who carry the HLA-DQ2 haplotype, influences the composition of the early gut microbiota, which could help determine the risk of CD. Similarly, other studies also reported a different gut microbiota composition (i.e., prevalence of *Bacteroides* species associated with intestinal inflammation) in high risk infants for the developmentof CD [26,27]. 

Some prospective cohort studies have examined the dynamics of the gut microbiota in genetically predisposed infants and have shown that alterations in the gut microbiota in the first few months (up to 12-24 months after birth) may contribute to the development of CD in this predisposed subgroup of children [43,44]. A recent multicenter study involving 31 infants from a large-scale prospective birth cohort study of infants with a first-degree relative with CD, examined the effects of genetic and environmental risk factors on the longitudinal development of the gut microbiota in this subset of patients, before the introduction of foods (including gluten) [28]. In details, the genetic risk to develop CD has been reported to be associated with a decreased abundance of several species of *Streptococcus* and *Coprococcus* at 4–6 months of age compared to those lacking genetic compatibility. The authors also reported that standard and high genetic risk for CD occurrence are associated with increased *Bacteroides* and *Enterococcus* species compared to no genetic risk, which is in line with previous data [25,27]; conversely, an association between genetic risk and an elevated amount of *Bifidobacterium* or *Proteobacteria*, already reported [25,27] was not found. Moreover, a decreased abundance of *Veillonella*, *Parabacteroides*, and *Clostridium perfringens* at 4–6 months after birth in infants with genetic compatibility was observed. The authors reported that, although the microbiome shifts observed in the first 6 months after birth increase the risk for the development of autoimmune diseases, including CD, it is still unclear whether they actually contribute to the later occurrence of CD.

Some perinatal factors including mode of delivery, type of infant feeding, and antibiotics have also been associated with an increased risk for CD development, and this may be due in part to their effects on the composition of the gut microbiota.

It has been reported that cesarean section may be associated with an increased risk of developing CD [10], however with inconsistent results so far [45]. Of note, infants born via cesarean section showed an increased amount of *Enterococcus faecalis* and a lower amount of *Bacteroides* and *Parabacteroides* compared to infants born vaginally [44], which might be relevant as *Parabacteroides* seem to decreasethe severity of intestinal inflammation [46].

There are no clear data on the influence of the type of feeding on the risk of developing CD. It has been reported that infant formula feeding is a risk factor for the development of CD [29], also due to its effects on the composition of the gut microbiota. However, according to other studies breastfeeding, on the other hand, has not been shown to be a protective factor [30]. According to a recent prospective study [28], including 31 infants with a first-degree relative with CD, any amount of milk formula (in both children fed exclusively on formula or in combination with breastmilk) was reported to be correlated with a decreased amount of *Bifidobacterium breve*, whereas exclusively formula fed infants demonstrated an increased amount of *Bifidobacteirum adolescentis*, *Ruminococcus gnavus* and *Lachnospiraceae*, being the latter associated to allergic disease in infants and increased intestinal inflammation [47].

The effects of early antibiotic exposure (including antibiotics given to the mother during delivery or to the infant at birth) on the composition of the microbiota is another factor that has been explored [28], although further studies are necessary to better understand the actual changes in the microbiota composition, that may precede the occurrence of CD, also considering that the available data are heterogeneous. In a cohort study, no statistically significant association was found between maternal antibiotic use during pregnancy and CD occurrence in offspring [31], which was confirmed by other authors [32,33]. On the other hand, some studies suggested a relationship between antibiotic use and CD [34], that may be dose-dependent [11,35]. 

Finally, acute gastrointestinal infections may affect the normal balance of the gut microbiota. Accordingly, some studies suggest that increased gastrointestinal infections (particularly *rotaviruses* and *enteroviruses*) in the first 6-18 months after birth are associated with an increased risk of CD [48], also by the resulting increased gut permeability. In details, there are heterogeneous data on a possible association with *rotaviruses*. According to Stene et al.,recurrent *rotavirus* infections were a predictor of a higher risk for CD autoimmunity [36], which was not confirmed in a later study that reported a 1.5% increase in CD prevalence over the last two decades, despite the introduction of *rotavirus* vaccine during the study period [37]. *Enteroviruses* [38], *Adenovirus type 12* [39], *Orthoreovirus* [40], and *Candida albicans* [41] have also been associated with an increased risk of CD. However, there are currently no studies examining the role of gastrointestinal infection on the gut microbiota in the specific subgroup of CD patients, so further research is needed to fully understand these mechanisms and develop targeted therapies for celiac disease.

### 1.3. Clinical and Therapeutic Implications 

The microbiome has been clearly shown to contribute to the pathophysiology of CD by modulating the immune response, regulating the integrity of the intestinal barrier and allowing the entry of gluten-derived proteins, and degrading gluten immunogenic peptides, even if there is no consensus about the specific microbial changes observed in this specific setting [49]. 

Several studies have identified different bacterial populations in CD patients when compared to healthy individuals. Of note, it is still not clear whether gut dysbiosis is the cause or the consequence of CD. Targeting the gut microbiota in CD patients is an attractive approach to promote the growth of beneficial bacteria and restore gut functions. Consistent with this hypothesis, novel therapies able to modulate the gut microbiome are being developed as complementary strategies [50] in CD. Indeed, probiotics, prebiotics, postbiotics, and fecal microbiota transplantation (FMT) together with the gluten-free diet, could play a role in modifying the gut microbiome, and maybe in the future, probiotics may be used as adjunct medications to reinforce/facilitate the gluten-free diet and improve symptoms [51]. It may be conceivable that restoring beneficial commensal species, such as obligate anaerobic producers of SCFA, may positively influence barrier function integrity and the host immune system. In addition, probiotics are an excellent source of endopeptidases that digest gluten [51], thus representing an adjuvant treatment option. 

There is recent evidence that several *lactobacilli* strains show the ability to hydrolyze immunogenic gluten peptides. Recently, a probiotic mix including *lactobacilli* and *bifidobacteria* strains has been reported to hydrolyze gluten peptides after the digestion of gliadin and to modify the pro-inflammatory state and the gliadin-induced epithelial modification in the gut [52]. FMT consists in a method by which donor gut microbiota is transferred into the digestive tract of the recipient, with the aim of restoring gut microbial imbalance towards eubiosis.The fecal matter of a healthy donor is delivered via either a nasogastric tube, colonoscope or capsule method [53]. According to available data, FMT is a safe procedure [54]. However, one systematic review reported an adverse event rate of 29%, the most frequentbeing abdominal discomfort, and up to 9.2% of serious adverse events occurred, even though these adverse events may not have been causally related to the FMT treatment itself but possibly to the disease process [55].

FMT has been widely used as a treatment for patients with recurrent *Clostridium difficile* (*C. dif*) infection [56] and has shown promise in the treatment of autoimmune conditions, including inflammatory bowel disease [53], where the microbiome may be implicated in the disease pathology. In the specific setting of CD, only few reports are available. Van Beurden et al. [57] reported a case of a patient with refractory CD type II who received FMT to treat *C. dif* infection and, surprisingly, showed a full recovery of duodenal villi and disappearance of CD-related symptoms. According to this report, one might speculate that speculate that modifying the composition of the gut microbiota could also help restore villous atrophy and, with this perspective, the role of FMT in treating diseases other than *C. dif* is exciting [58]. However, further studies are warranted to clarify the potential role and the safety of FMT in the treatment of CD, including the refractory forms.

Data from literature show that the several biological functions of the gut microbiota against immunomodulatory responses have made microbiome-based therapy an alternative therapeutic paradigm to improve both the symptoms related to CD and quality of life (Figure 1). 

However, the actual potential of microbiota-based techniques aimed at quantitatively and qualitatively altering the gut microbiota to treat and improve the symptoms of CD has yet to be fully evaluated and will be hopefully clarified by further research in the future [59].

## 2. Conclusions

The gut microbiota plays a complex role in the pathogenesis of CD. The gut microbiome is a complex community of microorganisms that can interact with the host immune system and modulate the response to gluten. Research has suggested that modifications in the composition and function of the gut microbiome may contribute to the development and progression of CD.

Several hypotheses have been proposed to explain the mechanisms by which the gut microbiome may contribute to CD pathogenesis. These include the expression of epitopes that mimic gluten, the activation of the immune system by lipopolysaccharides, increased inflammation caused by certain bacteria, and activation of the innate immune system by viral infections. 

Alterations in the gut microbiome, including a reduced abundance of beneficial bacteria, have also been reported to be associated with CD. In particular, strains of *Bifidobacterium* with potential anti-inflammatory properties, have been shown to be decreased in patients with CD. On the other hand, the overgrowth of certain bacteria, such as *Bacteroides* and *Escherichia Coli*, has been associated with increased intestinal permeability, which is a hallmark of CD. 

Moreover, the gut microbiome can influence the development and activation of the immune system, including the immune response to gluten in CD. 

Finally, the production of SCFAs by gut bacteria can also play a role in the regulation of intestinal barrier function and the immune response to dietary antigens. 

In conclusion, the gut microbiome is a complex and dynamic community that plays a crucial role in the development of CD. 

However, further research is needed to fully understand the complex interactions between the gut microbiome and the mechanisms by which the gut microbiome modulates the immune response to gluten.

Indeed, there are still some controversies as some authors reported no significant differences between the gut microbiota of CD patients compared to healthy subjects. Nonetheless, a deeper understanding of the role of the gut microbiota in CD pathogenesis might help in the prevention and also in the treatment of CD, leading to the development of microbiome-targeted therapies or microbiome-based complementary strategies, particularly in cases of complicated or refractory CD.

## Figures and Tables

**Figure 1 cells-12-00823-f001:**
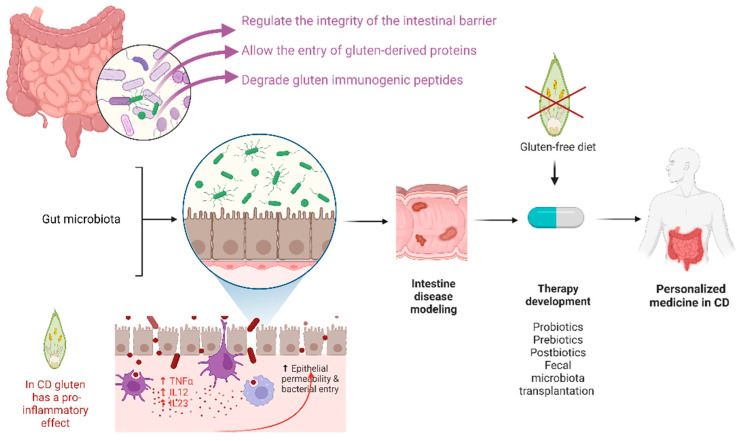
Biological functions of the gut microbiota with possible novel microbiome-based therapies under development as complementary strategies in celiac disease (CD).

**Table 1 cells-12-00823-t001:** Summary of the mechanisms by which the gut microbiota may participate in the pathogenesis of celiac disease (CD).

Mechanisms	Study
**Role of Environmental Factors**	
Bacterial expression of epitopes that mimic gliadin	Petersen [20]
Gram- negative bacterial lipopolysaccharide expression as a trigger for both the innate and adaptive immune systems	Kim [21]
Pseudomonas aeruginosa + gluten causes increased inflammation	Caminero [22]
Viral infections as a trigger for activation of the innate immune system	Araya [23]
**Role of Microbiota Itself**	
- Production of immunogenic/tolerogenic gluten peptides - Release of zonulin - Promotion of epithelial mucosal maturation. - Modulation of the immune system through the production of pro-inflammatory/anti-inflammatory cytokines and peptides	Cristofori [24]
**Role of Genetics**	
The genotype of infants at familial risk for CD influences the composition of the early gut microbiota	Olivares [25]
Infants genetically predisposed to CD show different gut microbiota composition (i.e., prevalence of *Bacteroides*)	Sanchez [26] De Palma [27]
Microbiome shifts observed in the first 6 months after birth in infants with a first-degree relative with CD can increase the risk for developing autoimmune diseases	Leonard [28]
**Role of Perinatal Factors**	
Cesarian section Higher risk of CD due to an increased amount of *Enterococcus faecalis* vs. a decreased abundance of *Bacteroides* and *Parabacteroides*	Leonard [28]
Feeding type - Infant formula feeding might be a risk factor for developing CD, but inconsistent results. - Exclusively formula fed infants showed an increased abundance of *Bifidobacteirum adolescentis*, *Ruminococcus gnavus* and *Lachnospiraceae*, being the latter associated with allergic disease in infants and increased intestinal inflammation	Akobeng [29] Szajewska [30] Leonard [28]
Antibiotic use - No significant association between maternal antibiotic use during pregnancy and CD in offspring - Dose-dependent association	Marild [31] Myleus [32] Kemppainen [33] Marild [34] Dydensborg Sander [35] Canova [11]
Gastrointestinal (GI) infections - GI infections during the first 6–18 months of life is associated with an increased risk of CD by enhanced permeability of the intestine - Inconsistent results for the association between *Rotaviruses* and risk of CD - Association between *Enteroviruses*, *Adenovirus type 12*, *Orthoreovirus*, *Candida albicans* and risk of CD	Marild [34] Stene [36] Gatti [37] Lindfors [38] Lahdeaho [39] Bouziat [40] Corouge [41]

## Data Availability

Not applicable. Data sharing is not applicable to this article as no new data were created or analyzed in this study.

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
