# Peer review of "Interaction between Gut Microbiota and Celiac Disease: From Pathogenesis to Treatment"

_cells, 2023, doi:10.3390/cells12060823_

Round 1

Reviewer 1 Report

 Celiac disease (CD) i9s one of auto-immune disease with intestinal microbiome influences.

Authors  clearly describing nature of CD diseases . To follow the mechanisms  of CDs figure presenting proinflammatory action of gluten will be helpful (chapter 1.1. Pathogenesis).

The chapter of microbiome role is less than  CD pathogenesis clear cut. Did Authors conclude that only single microbiota bacterial species are sufficient to potentiate of induce CD?  The missing part of manuscript  is comparison of non-CD microbiota subjects with the similar on sex and ages subjects. May be shift on abundance of Protobacteria (with pro-inflammatory potencies)  might tiger CD? .

They are ample of data that lactic bacteria species by producing  short chain organic acids i.e. butanol, acetic, propionic acids  acting as anti-inflammatory agents. That information are missing.

Minor correction – bacterial species names generally are presented in italic forms,

 Celiac disease (CD) i9s one of auto-immune disease with intestinal microbiome influences.

Authors  clearly describing nature of CD diseases . To follow the mechanisms  of CDs figure presenting proinflammatory action of gluten will be helpful (chapter 1.1. Pathogenesis).

The chapter of microbiome role is less than  CD pathogenesis clear cut. Did Authors conclude that only single microbiota bacterial species are sufficient to potentiate of induce CD?  The missing part of manuscript  is comparison of non-CD microbiota subjects with the similar on sex and ages subjects. May be shift on abundance of Protobacteria (with pro-inflammatory potencies)  might tiger CD? .

They are ample of data that lactic bacteria species by producing  short chain organic acids i.e. butanol, acetic, propionic acids  acting as anti-inflammatory agents. That information are missing.

Minor correction – bacterial species names generally are presented in italic forms,

 Celiac disease (CD) i9s one of auto-immune disease with intestinal microbiome influences.

Authors  clearly describing nature of CD diseases . To follow the mechanisms  of CDs figure presenting proinflammatory action of gluten will be helpful (chapter 1.1. Pathogenesis).

The chapter of microbiome role is less than  CD pathogenesis clear cut. Did Authors conclude that only single microbiota bacterial species are sufficient to potentiate of induce CD?  The missing part of manuscript  is comparison of non-CD microbiota subjects with the similar on sex and ages subjects. May be shift on abundance of Protobacteria (with pro-inflammatory potencies)  might tiger CD? .

They are ample of data that lactic bacteria species by producing  short chain organic acids i.e. butanol, acetic, propionic acids  acting as anti-inflammatory agents. That information are missing.

Minor correction – bacterial species names generally are presented in italic forms,

Author Response

Reviewer 1

Celiac disease (CD) is one of auto-immune disease with intestinal microbiome influences.

Authors clearly describing nature of CD diseases. To follow the mechanisms of CDs figure presenting proinflammatory action of gluten will be helpful (chapter 1.1. Pathogenesis).

As suggested by the Reviewer, Figure 1 has been edited with the data regarding proinflammatory action of gluten.

The chapter of microbiome role is less than CD pathogenesis clear cut. Did Authors conclude that only single microbiota bacterial species are sufficient to potentiate of induce CD?  The missing part of manuscript is comparison of non-CD microbiota subjects with the similar on sex and ages subjects. May be shift on abundance of Protobacteria (with pro-inflammatory potencies) might tiger CD? .

We thank the Reviewer for these interesting observations. According to available data, patients at risk of developing CD showed alterations in the gut microbiota, but a single bacterial species was not identified. As reported in the text “Some prospective cohort studies have also examined the dynamics of the gut micro-biota in genetically susceptible infants and have shown that changes in the gut microbiota in the first few months (up to 12-24 months after birth) may contribute to the development of CD in this predisposed subgroup of children [28,29]. A recent multicenter study involving 31 infants from a large-scale prospective birth cohort study of infants with a first-degree relative with CD, examined the effects of genetic and environmental risk factors on the longitudinal development of the gut microbiota in this subset of patients, before the introduction of foods (including gluten) [30]. The authors reported that, although the microbiome shifts observed in the first 6 months after birth increase the risk for developing autoimmune diseases, including CD, it is still unclear whether they actually contribute to the later development of CD” even if it appears that the microbiome shifts observed in the first 6 months after birth increase the risk for developing CD, it is not clarified whether they actually contribute to the later development of CD. As suggested, we have added more specific details regarding the bacterial species associated with an increased risk of developing CD (Page 5, lines 160-169): “In details, both high and standard genetic risk to develop CD have been reported to be associated with a decreased abundance of several species of Streptococcus and Coprococcus at 4–6 months of age compared to those lacking genetic compatibility. The authors also reported that standard and high genetic risk for developing CD are associated with an increased abundance of Bacteroides and Enterococcus species compared to no genetic risk, which is in line with previous data [25, 27]; conversely, an association between genetic risk and increased amount of Bifidobacterium or Proteobacteria, already reported [25,27] was not found. Moreover, a decreased abundance of Veillonella, Parabacteroides, and Clostridium perfringens at 4–6 months after birth in infants with high and standard genetic compatibility was observed”.

They are ample of data that lactic bacteria species by producing short chain organic acids i.e. butanol, acetic, propionic acids acting as anti-inflammatory agents. That information are missing.

As suggested, a pertinent comment has been added (page 6, lines 230-232 and 235-239, reference: Giorgi et al. Nutrients 2020).

Minor correction – bacterial species names generally are presented in italic forms,

As suggested, bacterial species names have been reported in italic forms.

Reviewer 2 Report

Celiac disease (CD) is a chronic, multiple-organ autoimmune disorder primarily affecting the small intestine induced by the ingestion of gluten. CD affects 1–2% of the general population and appears in genetically predisposed people of all ages, but gut microbiota still plays a significant role in CD. In this review, the authors focused on the correlation between microbiota and CD. Exposure to gluten is essential for the development of CD, and the lower abundance of beneficial bacteria in the microbiome is another crucial factor to promote CD. Mechanisms, such as genetics, perinatal factors, and environmental factors, have been introduced in a table. Finally, the authors concluded the best treatment for CD were probiotics, prebiotics, postbiotics, and fecal microbiota transplantation with a gluten-free diet. It is an interesting review; However, I still have a little concern regarding the review.

 1.      Fecal microbiota transplantation (FMT), a stool transplant, is a new protocol tried to treat some gut diseases recently. It has brought some benefits to patients, but it still brings some side effects. Please introduce the role of FMT in the CD in detail. 

Author Response

Reviewer 2

Celiac disease (CD) is a chronic, multiple-organ autoimmune disorder primarily affecting the small intestine induced by the ingestion of gluten. CD affects 1–2% of the general population and appears in genetically predisposed people of all ages, but gut microbiota still plays a significant role in CD. In this review, the authors focused on the correlation between microbiota and CD. Exposure to gluten is essential for the development of CD, and the lower abundance of beneficial bacteria in the microbiome is another crucial factor to promote CD. Mechanisms, such as genetics, perinatal factors, and environmental factors, have been introduced in a table. Finally, the authors concluded the best treatment for CD were probiotics, prebiotics, postbiotics, and fecal microbiota transplantation with a gluten-free diet. It is an interesting review; However, I still have a little concern regarding the review.

  1. Fecal microbiota transplantation (FMT), a stool transplant, is a new protocol tried to treat some gut diseases recently. It has brought some benefits to patients, but it still brings some side effects. Please introduce the role of FMT in the CD in detail. 

We thank the reviewer for the positive comment and this interesting observation. Accordingly, we have added some specific comments regarding the potential and interesting role of FMT in celiac disease; we have also described possible related side effects (see page 6 and 7, references: Liptak et al. Med. Hypotheses 2019; Al et al. Mult Scler J Exp Transl Clin 2022; Wang et al. PLoS ONE. 2016; Belvoncikova P, et al. Int J Mol Sci. 2022; van Beurden et al. J Gastrointestin Liver Dis 2016; Golfeyz S, Am J Gastroenterol. 2018).